# Rotation of 3D Anatomy Models Is Associated with Underperformance of Students with Low Visual-Spatial Abilities: A Two-Center Randomized Crossover Trial

Bo S. van Leeuwen [1,*], Anna E. D. Dollé [1], Johannes C. M. Vernooij [2], Beerend P. Hierck [1,†] and Daniela C. F. Salvatori [1,†]

1 Anatomy and Physiology, Department of Clinical Sciences, Faculty of Veterinary Medicine, Utrecht University, Yalelaan 1, 3584 CL Utrecht, The Netherlands; a.e.d.dolle@uu.nl (A.E.D.D.); b.p.hierck@uu.nl (B.P.H.); d.salvatori@uu.nl (D.C.F.S.)

2 Department of Population Health Sciences, Faculty of Veterinary Medicine, Utrecht University, Yalelaan 7, 3584 CL Utrecht, The Netherlands; j.c.m.vernooij@uu.nl

* Correspondence: b.s.vanleeuwen@uu.nl

† These authors contributed equally to this work.

**Abstract:** Virtual 3D models can be an animal-free alternative to cadaveric dissection to learn spatial anatomy. The aim of this study was to investigate if the learning outcome differs when studying 3D models with a 360° rotatable interactive monoscopic 3-dimensional (iM3D) or an interactive monoscopic 2-dimensional (iM2D) visualization, and whether the level of visual-spatial ability (VSA) influences learning outcome. A two-center randomized crossover trial was conducted during the Laboratory Animals Science Course (March 2021–March 2022). Participants studied a 3D rat model using iM3D and iM2D. VSA was assessed by a 24-item mental rotation test and learning outcome by two knowledge tests. Data from 69 out of 111 recruited participants were analyzed using linear regression. Participants with low VSA performed significantly worse compared to participants with medium or high VSA when using iM3D, but equally well when using iM2D. When VSA level was disregarded, participants performed equally well with both visualizations. Rotation in iM3D requires the student to construct a mental 3D image from multiple views. This presumably increases cognitive load, especially for students with low VSA who might become cognitively overloaded. Future research could focus on adapting the visualization technique to students' personal needs and abilities.

**Keywords:** visual-spatial ability; animal-free anatomy education; veterinary; monoscopic; mental rotation; two-dimensional display

## 1. Introduction

Since anatomical knowledge is essential for students to appreciate and comprehend other subjects, such as pathology, radiology, and surgery, animal anatomy can be considered a cornerstone of veterinary and biomedical education. After graduation, adequate knowledge of anatomy plays a continuously interwoven part in many aspects of clinical practice [1], and comparative anatomy is essential for researchers who work with animal models [2,3].

Traditionally, cadaveric dissection has been the teaching method of choice for students to thoroughly understand spatial anatomy [4,5]. However, time available for this labour-intensive and expensive type of education is diminishing. There are also other problems associated with cadaveric dissections, including the storage and fixation of cadavers, which involves the widely applied use of formaldehyde, a chemical that it is expected to soon be banned because of the health hazard, as an airway, skin, and eye irritant and being carcinogenic to humans [6]. Moreover, some studies mention other disadvantages of harmful animal use in education, such as distraction from relevant scientific concepts by the plight of the animals [7] or the psychological impact on students [8]. Educational experiences perceived as morally wrong might lead to desensitization [9] and compassion fatigue [10].

Focusing on European veterinary medicine and life sciences education, the use of laboratory animals needs to conform with the EU directive 2010/63 [11], which legally obliges educators and researchers to introduce, whenever possible, alternatives to the educational use of animals. Unfortunately, this requirement is often not complied with, as demonstrated by the constantly high numbers in the EU statistics of animal use for educational purposes [12,13]. There is also a strong ethical and societal movement advocating for animal-free education and research purposes [14]. Possible solutions include the use of digital anatomy models and, where necessary, using ethically sourced animals, i.e., available through a donation program [15], preferably as plastinated specimens to extend their shelf life [16].

Several reviews have shown that teaching without harmful animal use leads to equivalent or even better learning outcomes [17–19]. With continuously increasing computer power, which is easily accessible using home devices and so also for education purposes, the use of digital (anatomy) models has become increasingly feasible [20]. Nowadays, digital models are three-dimensional (3D), and can be visualized and manipulated from a computer/tablet/cell phone screen or through head-mounted devices [20–22]. Although these models offer an animal-free solution for the interactive perception of spatial anatomy, are they sufficient to train, e.g., surgeons to perform a complex procedure on a live animal or human? Research into this field is gradually emerging, and although many studies show positive results, others conclude that technology is not ready to support or replace learning in an evidence-based manner [23,24]. One of the many aspects that directs the efficacy of learning spatial anatomy from digital models is the projection method of the digital content [25]. When a 3D model is displayed on a flat medium, like a regular computer/tablet/cell phone screen, the human eyes perceive this as 2D information and store it as such [26]. Transferring this 2D knowledge into the 3D domain requires a considerable level of visual-spatial ability (VSA), i.e., the ability to mentally reconstruct and rotate 2D into 3D information [27,28]. We and others have described that when learning spatial anatomical relationships from rotating 3D models projected on 2D screens, students with below-average VSA perform significantly less well than those with above-average VSA [29–31]. At the same time, they perform equally well when allowed to learn from exactly the same models in a digital 3D learning environment, i.e., in Augmented Reality (AR) or Virtual Reality (VR), where the model is projected in 3D [29]. Garg et al. have suggested that key views of a 3D object could be beneficial to this group compared to rotation views [32]. This means that projection technology is an important and determining factor in the learning process when it comes to digital models.

Three-dimensional projection technologies, e.g., in VR and AR, make use of the disparity between the human eyes to perceive depth: the left and right eyes are presented with a slightly shifted and rotated image which simulates our natural vision [33]. In addition, extra 3D information can be added by dynamic exploration, which, for example in AR/VR, means that you can physically approach the virtual model and rotate yourself around the model [34]. This makes AR/VR technology very interesting for learning spatial relationships.

When both eyes look at the same 2D image, vision is essentially monoscopic, even if the image is from a 3D model, as described above. In the literature, this is called monoscopic 3D [35]. In this case, a three-dimensional shape is perceived through monocular cues, such as interposition, texture gradients, contrast and shadows [36]. Software enables different types of interaction to change the presentation of the virtual 3D model in 2D space. This allows for the perception of extra spatial information through monocular cues. For example, removing an anatomical structure makes a new structure visible, giving the impression that the former was on top of the latter. We call this interactive monoscopic 2D (iM2D). When the visualization also allows for the rotation of the 3D model, thereby changing the viewpoint, it is called interactive monoscopic 3D (iM3D). Recent systematic reviews [35,37] show that the use of these interactive monoscopic visualizations to display the 3D models lead to higher learning outcomes compared to non-interactive visualizations, and that the learning outcome depends on the level of VSA of the learner [35]. The cognitive load or

mental effort of the learner probably varies depending on the level of VSA and the type of interaction (e.g., removing layers or rotation).

Building on this paradigm, this study aims to investigate if the learning outcome is different when studying spatial anatomical relationships with an iM3D or an iM2D visualization, and whether the level of VSA interacts with the visualization and influence learning outcome. The study was embedded in the Laboratory Animal Science (LAS) course of two Dutch Universities (Utrecht and Leiden Universities). The LAS courses are for the qualification of scientists and students who need to use laboratory animals for scientific purposes. In the Netherlands alone, around 500.000 laboratory animals per year are used in science and education [38]. In the education domain, the highest number of animals is used in the LAS courses [14]. The LAS courses, which are organized by almost all Dutch Universities, account for approximately 3300 laboratory animals per year. Most of the LAS courses in The Netherlands are organized as postgraduate education, but in a few specific cases, they are also as part of a Master's program. There is the ambition to abolish obligatory LAS courses at the Master's level since most students in this phase of their career have not yet decided about their future [14]. The LAS courses consist of a broad theoretical basis and a practical part, usually using live animals, which together lead to the required certification. The suggestion is to modernize the LAS courses by expanding the theoretical part and possibly including animal-free innovations (thereby also making this course more relevant to all biomedical Master's students), resulting in a new format that does not use live laboratory animals at all. The practical part should be made available as a more personalized course, i.e., only for those who are going to perform actual animal experiments [14]. Exploring the efficacy of animal-free methods for learning anatomy for this group of students and future professionals is therefore relevant.

## 2. Materials and Methods

### 2.1. Interactive Monoscopic Two-Dimensional and Three-Dimensional Visualisation

A 3D model of a rat [39] was annotated and corrected for anatomical structures using Blender [40]. Subsets of key anatomical structures of the digestive region of a female rat and the urogenital region of a male rat were exported to a babylon file format using BlenderExporter [41]. For the iM2D and iM3D visualizations in the web browser, a Babylon.js web viewer [42] was customized.

The iM2D visualization allowed the user to interact with the 3D model and to change the presentation by hiding and showing individual anatomical structures, displaying names and highlighting structures when clicked upon (see Figure 1A). The ventral viewpoint was fixed and could not be changed in iM2D. The ventral viewpoint was chosen because it is the most common point of view when performing a cadaveric dissection on a laboratory rat.

The iM3D visualization offered the interaction as previously described for the iM2D visualization and additionally allowed for the 360-degree rotation of the 3D model, as well as panning motion and zooming to change the initial ventral point of view (see Figure 1B).

### 2.2. Visual-Spatial Ability Test

Visual-spatial abilities of the participants were evaluated by the 24-item Mental Rotation Test (MRT) presented by Shepard and Metzler (1971) [43], which was previously validated by Vandenberg and Kuse (1978) [44] and redrawn by Peters et al. (1995) [45]. Participants could score 1 point per item by selecting the two rotation variants of the target figure from four possible choices. In all other cases, no points were awarded for that item. The total score was calculated by the sum of all awarded points; thus, a maximum score of 24 points could be obtained. Prior to taking the MRT, participants were provided with test instructions and an example item with no time limit. Subsequently, the participants were given 10 min to complete the MRT, in which they were not allowed to return to previous items once their answers were submitted.

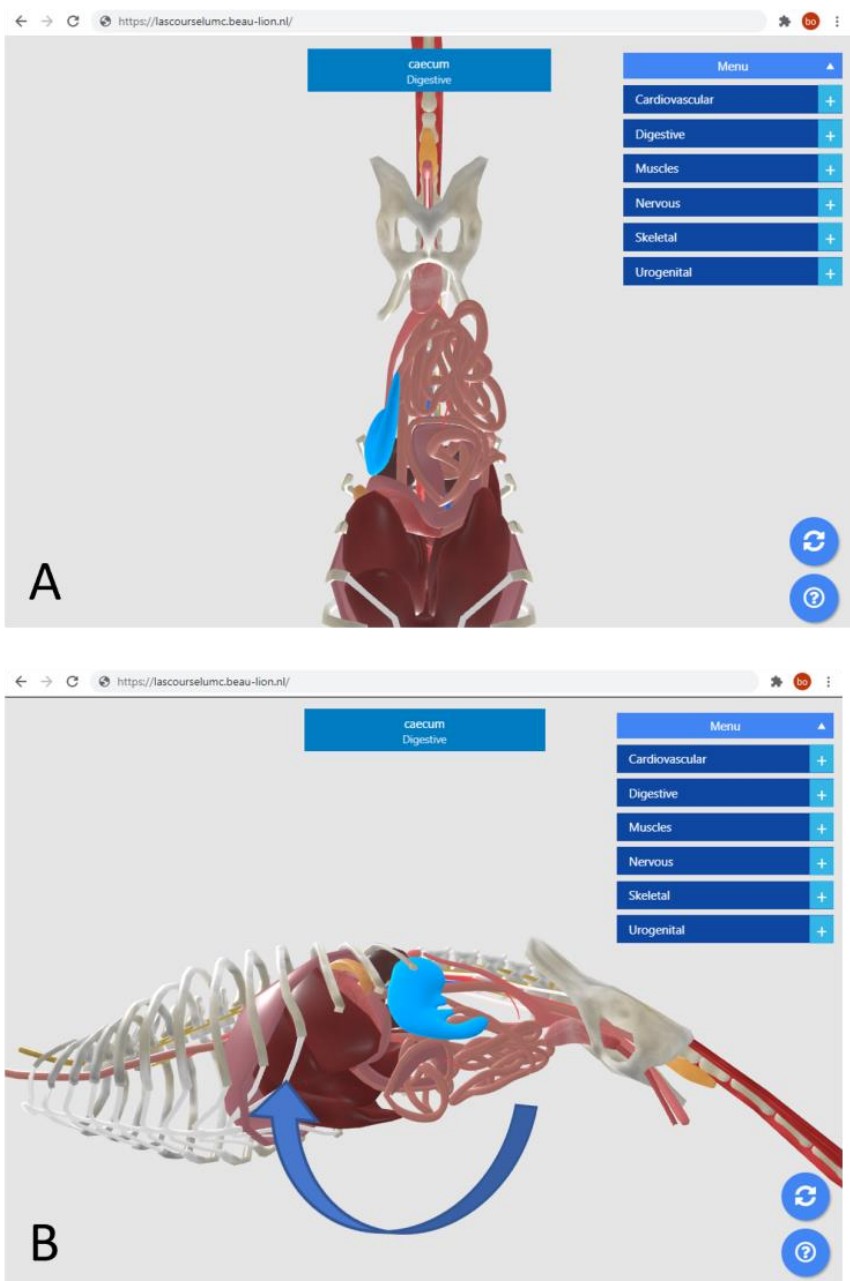

**Figure 1.** iM2D and iM3D visualizations. (**A**) Participants can select and show/hide structures when the anatomical model of the rat is visualized in iM2D, but they cannot change the ventral point of view of the model and vertical orientation. (**B**) When participants are presented with the anatomical model of the rat in iM3D visualization, they have the additional functionality to zoom and pan the model and to rotate the model 360 degrees in all directions to change their point of view and orientation (illustrated by the blue bended arrow and the changed orientation of the model).

### 2.3. Study Design

A two-center randomized crossover trial was carried out during six LAS courses at the Leiden University Medical Center (LUMC) and four LAS courses at the Utrecht University (UU) faculty of veterinary medicine from March 2021 to March 2022. A crossover design ensured that all participants could interact with both visualizations and allowed for a within-subject comparison of the effect of the visualization on the learning outcome (Figure 2). Participants of group A started with iM3D followed by iM2D and vice versa for the participants of group B. Participation was voluntary, and written consent was obtained

from all participants. The study protocol was approved by the local course coordinators and received ethical approval from the Netherlands Association of Medical Education (NVMO) (NERB 2021.6.9).

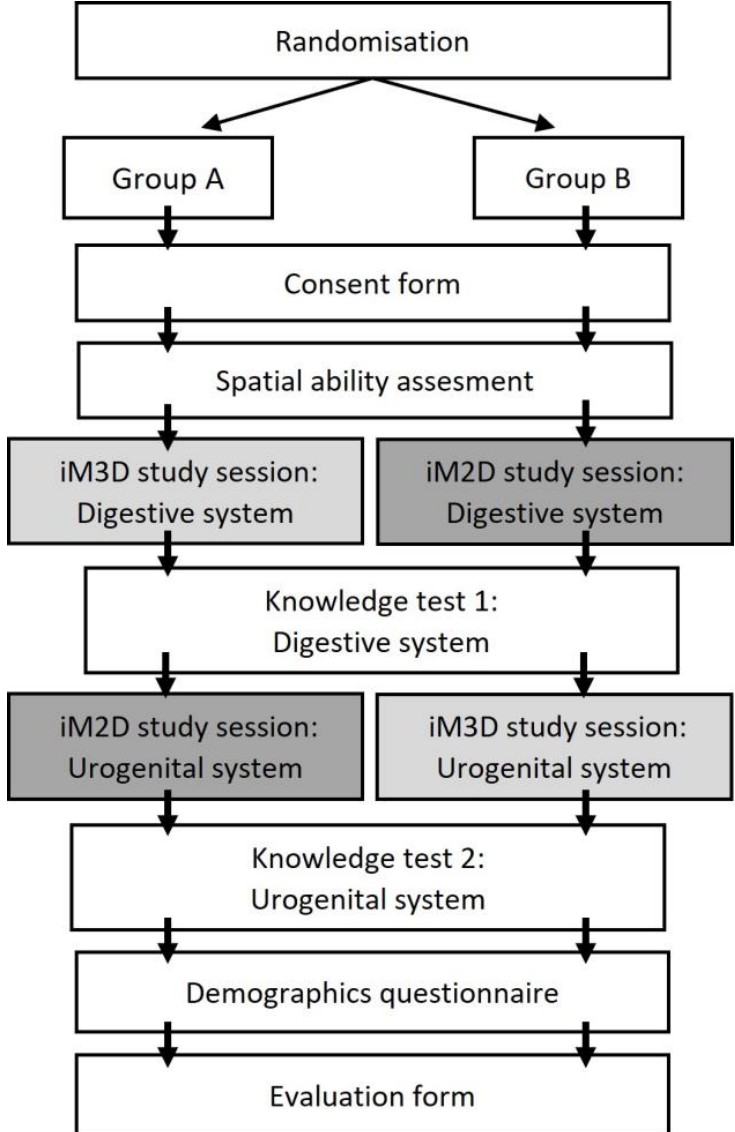

**Figure 2.** Flowchart of the crossover study design. After randomization and obtained consent, participants' visual-spatial ability was assessed with a mental rotation test. After this, participants were offered a 10 min study session with the digestive system of the female rat model either presented in interactive monoscopic three-dimensional (iM3D) or interactive monoscopic two-dimensional (iM2D) visualization. Participants' acquired knowledge was assessed with an anatomical knowledge test, focusing on the acquisition of spatial knowledge. Next, participants received another 10 min study session with the urogenital system of the male rat model. Participants that used the iM3D visualization previously, were now presented with the iM2D visualization, and vice versa. Again, participants' acquired knowledge was assessed with an anatomical knowledge test. Finally, participants' characteristics and learning experiences were obtained through a questionnaire.

*2.4. Study Population*

Eligible participants were students enrolled in the LAS course either at the LUMC or at the UU at the time an experiment was scheduled. As an entrance requirement for the LAS course, all students had to pass an obligatory entry test to demonstrate their knowledge of basic vertebrate anatomy, zoology, and physiology.

The LAS UU and LUMC courses focus on presenting basic facts and principles that are essential for the humane use and care of animals used for scientific purposes and for the quality of research. The courses have been recognized for the qualification of scientists who wish to use animals for scientific purposes, in compliance with the Dutch Experiments on Animal Act by the Dutch Competent Authority [46]. The UU [47] and LUMC courses [48] have been accredited by the Federation of European Laboratory Animal Science.

Participants were recruited for the study through an information email in the week prior to the beginning of the LAS course and a 10 min informative lecture by one of the researchers on day 1 of the course. The experiment was conducted between day 1 and day 3 of the course, before the scheduled anatomy lecture to ensure limited anatomy knowledge acquisition before the experiment. The participation did not affect students' final grades in any way and did not interfere with the LAS course content. Participating students received no compensation to take part in the research study.

## 2.5. Randomisation and Pseudonymisation

The students that applied received confirmation of their participation via email. A custom-made excel randomizer was used by the first author, who was also involved in the data analysis, to approach an equal number of participants in Group A and B for every scheduled experiment (Cohort). The participants were blinded to the group they were assigned to. Pseudonymization was performed by assigning a random four-digit number to each participant as an identifier (pseudoID). The record linking the pseudoID to a specific person was stored separately from the rest of the data and was not used for further analysis.

## 2.6. Study Sessions and Knowledge Tests

Two sets of specific learning goals were presented to the participants concerning the digestive system and male urogenital system of the rat (Appendix A). The participants were presented with the iM2D visualization or the iM3D visualization, depending on the group the participant was appointed to. Participants had 10 min of study time per topic. Learning outcomes were evaluated by a knowledge test directly following each study session. Both tests contained four questions tailored to the learning goals of the specific organ system (Appendix B) and designed to examine the spatial knowledge that students acquired. Participants were given 10 min to complete each test and were not allowed to consult the visualizations during the test. They could review and alter all answers within the time slot before final submission. Participants could score 2 points per question; thus, a maximum score of 8 points could be obtained per knowledge test. The knowledge test score was converted to a percentage of the maximum score (0–100%).

## 2.7. Participant's Demographics

Gender was measured in three categories, namely female (F), male (M), and unknown, depending on their self-reported data. Age was measured on a continuous scale in years. Participants could indicate their previous education through an open-ended question. Previous education was categorized into "Bachelor of Science degree or lower" (≤BSc) and "Master of Science degree or higher" (≥MSc).

## 2.8. Evaluation of Learning Experience

The perceived learning experience with the iM3D and iM2D visualizations was evaluated with a questionnaire (Appendix C). Participants could score statements on a 5-point Likert scale (1 = strongly disagree; 2 = disagree; 3 = neither agree nor disagree; 4 = agree; 5 = strongly agree). Furthermore, participants were given the opportunity to give feedback on the experiment and report technical issues.

## 2.9. Experimental Environment

Qualtrics Survey software (Qualtrics, version 1.2021, Provo, UT, USA) was used to design the digital experimental environment, which consisted of a consent form, mental rotation test

(MRT), two study sessions with the visualization (iM3D/iM2D) for that specific study session, two knowledge tests, and a questionnaire concerning the participant's demographics and evaluation of the learning experience. The survey software also managed the time restrictions on the MRT test, study sessions, and knowledge tests. A personal link to the experimental environment was generated through Qualtrics and e-mailed to the participants. In addition, participants received an invitation to a Microsoft Teams environment for a standardized 5 min walkthrough by the first author on the use of the webviewer and the interactions with the anatomical model in iM2D and iM3D visualizations. After the walkthrough, participants could use the Microsoft Teams environment for technical support.

*2.10. Statistical Analysis*

Statistical analysis was performed using R (version 4.1.3) [49] within RStudio [50]. Participants were excluded from the dataset in the case of missing demographic data, reported technical issues, 0 points on the MRT, or 0 points on the knowledge tests. Participants' baseline characteristics were summarized using descriptive statistics. The differences in distribution of Gender and Previous Education between groups and between locations were assessed with Pearson's $X^2$ tests and for Age and MRT with the Wilcoxon rank sum test with continuity correction.

To determine the effects of the independent variables on the scores of the knowledge tests, a multivariable mixed-effects regression analysis was performed using the R package nlme [51]. In this analysis, the pseudoID was taken as a random effect to account for the repeated measurements within the participant, and visualization (iM2D or iM3D), MRT score (continuous scale), Group (A or B), Cohort (1–10), Gender (male/female/unknown), Age (continuous scale) and Previous Education ($\leq$BSc/$\geq$MSc) as fixed effects. Furthermore, three interaction terms (visualization*group, visualization*MRT, visualization*age) were added to the model to test for possible interactions. QQ-plots and residuals versus predicted values plots were used to check for normality and homoscedasticity of the full model. The assumption of linearity with MRT was not fulfilled, and therefore, we categorized MRT in three categories, namely low, medium, and high, based on the tertile distribution. Backward selection based on Akaike's Information Criterion (AIC) was used for model reduction for the fixed part. A parameter was dropped from the analysis if it resulted in an equal (i.e., an increase of +2 was considered as equal fit for the parsimonious model) or decreased AIC value. Backward selection started with the interaction terms, followed by the main independent variables if not part of remaining interactions. The variable visualization was not dropped for reduction regardless of the AIC because this term was essential to answer our research questions.

The results of the final model was presented as estimates of the regression coefficients or differences between group means with a 95% confidence interval (CI).

**3. Results**

A total number of 111 participants was recruited for this study from six LAS courses at the LUMC and five courses at the UU between March 2021 and March 2022. All of participants were randomly allocated to group A or B (see Figure 3 for the participant flow). Participants were blinded to their sequence allocation prior to the experiment. Participants who did not show up (n = 17), did not finish the entire survey (n = 15), scored 0 points on the MRT (n = 1), or scored 0 points on the knowledge test (n = 6) were excluded from the analysis. Two participants were excluded because they reported technical issues during the experiment, resulting in the inability to view the anatomical model in the webviewer. One participant had incomplete data and was therefore excluded. This resulted in a total of 69 participants that were included in the final analysis. The distribution of the demographic characteristics and the VSA of the included participants were similar between group A and B (Table 1), and academic institutions (Appendix D).

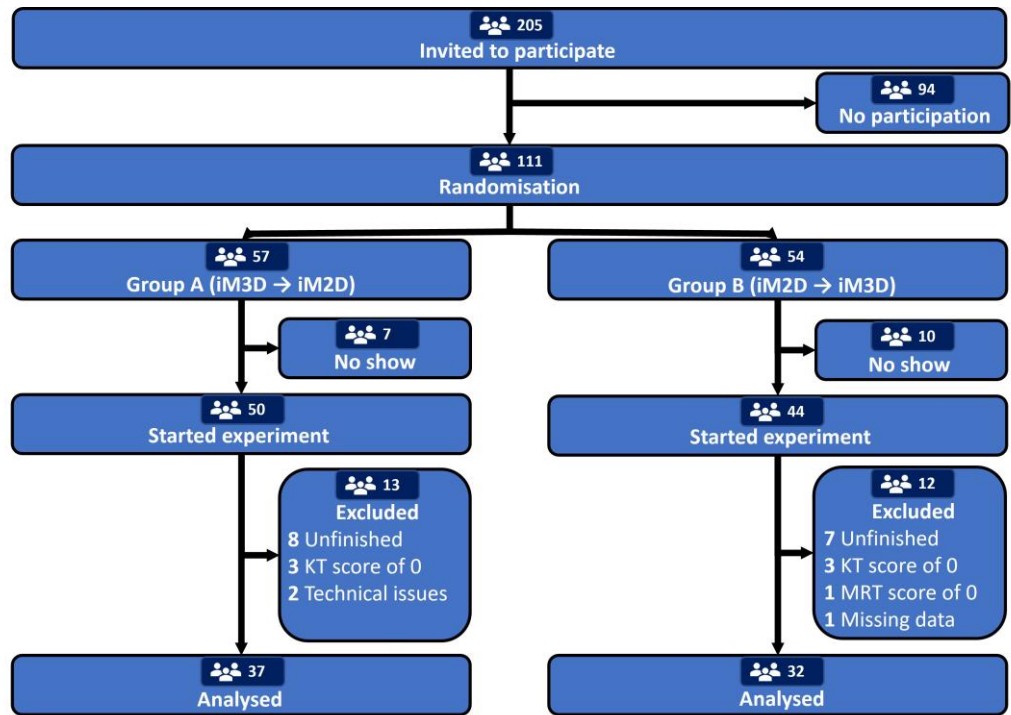

**Figure 3.** Participants flowchart. In total, 111 Participants entered the experiment and were randomised for group A or B. Indicated dropouts resulted in 37 data points in group A vs. 32 data points in group B. iM3D, Interactive monoscopic three-dimensional; iM2D, interactive monoscopic two-dimensional; KT, knowledge test; MRT, mental rotation test.

**Table 1.** Baseline demographic and visual-spatial ability characteristics by group and by total.

| Characteristic | Group A iM3D→iM2D (n = 37) | Group B iM2D→iM3D (n = 32) | Total (n = 69) | *p*-Value |
|---|---|---|---|---|
| Academic Institute, n (%) | | | | |
| Leiden University Medical Center | 26 (70) | 23 (72) | 49 (71) | 0.88 |
| Utrecht University | 11 (30) | 9 (28) | 20 (29) | |
| Gender, n (%) | | | | |
| Female | 27 (73) | 20 (62) | 47 (68.1) | |
| Male | 10 (27) | 11 (34) | 21 (30.4) | 0.42 |
| Unknown | 0 | 1 (3) | 1 (1.4) | |
| Age, mean (±SD) | 25.8 (4.7) | 26.5 (4.7) | 26.1 (4.7) | 0.63 |
| Previous Education, n (%) | | | | |
| ≤BSc | 14 (38) | 16 (50) | 30 (43.5) | 0.31 |
| ≥MSc | 23 (62) | 16 (50) | 39 (56.5) | |
| Mental Rotation Test, mean (±SD) | 15.6 (5.3) | 16.5 (5.6) | 16.0 (5.4) | 0.39 |

*p*-values for the difference in distribution of Gender, Previous Education, and Academic Institute between groups were calculated using Pearson's $X^2$ tests and for Age and MRT using the Wilcoxon rank sum test with continuity correction. BSc = Bachelor of Science degree; iM3D = interactive monoscopic three-dimensional; iM2D = interactive monoscopic two-dimensional; MRT = mental rotation test; MSc = Master of Science degree; SD = standard deviation.

### 3.1. Overall Scores on the Anatomy Knowledge Test

Participants scored equally well on the anatomy knowledge test after using the iM2D and iM3D visualizations (Figure 4). With an average score of 44.3% (SD = 17.3) after using

the iM2D visualization compared to 47.0% (SD = 19.4) (β = 2.7%, 95% CI [−3.0, 8.4], *p* = 0.35) after using the iM3D visualization.

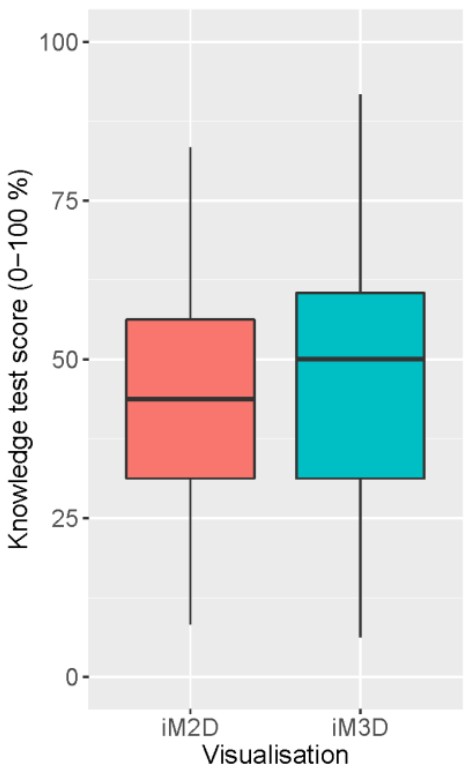

**Figure 4.** Knowledge test scores related to iM2D and iM3D. Participants mean knowledge test scores were not significantly better after studying with the interactive monoscopic three-dimensional (iM3D) visualization (47.0%, SD ± 19.4) compared to the interactive monoscopic two-dimensional (iM2D) visualization (44.3%, SD ± 17.3; *p* = 0.35).

### *3.2. Effects of Study Parameters on Knowledge Test Scores*

To further study the effect of the MRT score on the learning outcome, participants were grouped by VSA level based on their MRT scores (MRT ≤ 13 = low, MRT 14–19 = medium, MRT 20–24 = high). Figure 5 shows that participants with high and medium VSA had comparable learning outcomes, whereas participants with low VSA significantly underperformed with iM3D visualization (β = −14.5%, 95% CI [−24.5, −4.5], *p* = 0.006), but not with iM2D visualization (β = −3.7%, 95% CI [−13.8, 6.3], *p* = 0.47). The other independent variables or interaction terms did not have a significant effect on the result of the knowledge test.

### *3.3. Participants Feedback*

On average, participants reported that the iM3D visualization allowed them to better understand the location of organs (mean = 4.3, SD = 0.8) and the spatial relationships between anatomical structures in the body (mean = 4.1, SD = 0.8). The ability to rotate the anatomical model in the iM3D visualization was considered useful (mean = 4.6, SD = 0.6). Interestingly, participants with low VSA responded equally positive compared to students with higher VSA (Table 2). Most participants with low VSA preferred the iM3D over the iM2D visualization (21 out of 25) and thought they benefitted from it. This opinion was also reflected in the open-text responses, in which participants with low VSA commented positive on the ability to rotate the model in the iM3D visualization, stating, for example, that "It was nice to be able to rotate the animal" and "I preferred that you could roll through the model and were able to view it from different sides".

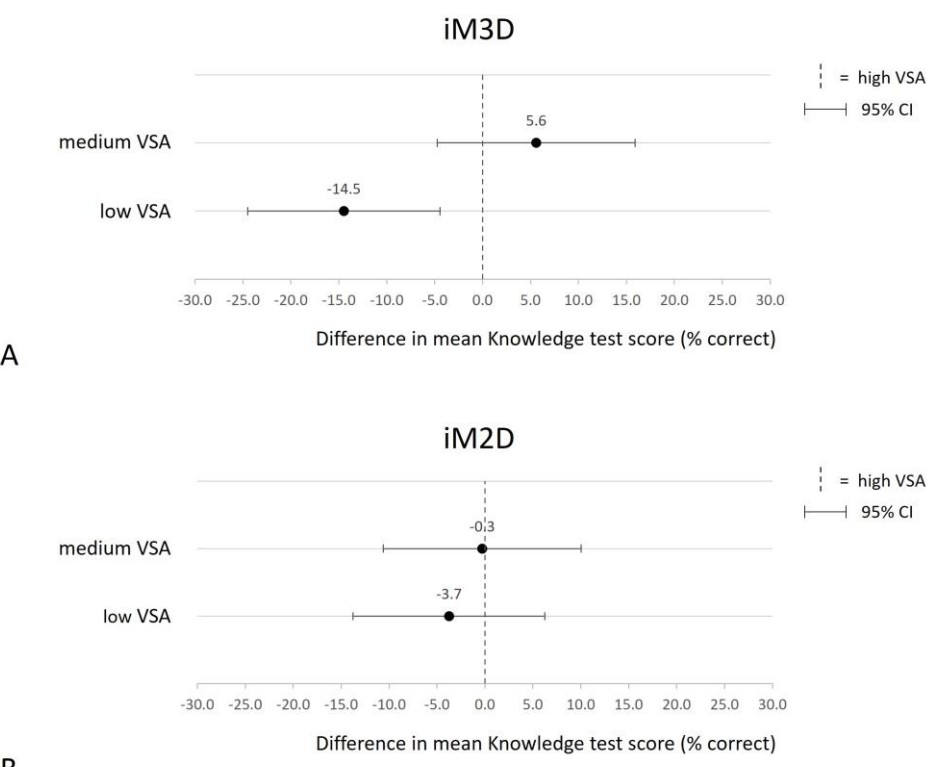

**Figure 5.** Visual-spatial ability influences the knowledge test score outcome after using iM3D or iM2D. Differences in mean percentage correct answers on the knowledge test score using the iM3D or iM2D visualizations, grouped by participants visual-spatial ability (MRT $\leq$ 13 = low, MRT 14–19 = medium, MRT 20–24 = high). The vertical dashed line represents the performance of the participants in the high VSA category as a reference. Participants with low or medium VSA that used an iM3D model had a $-14.5\%$ (CI 95% [$-24.5$, $-4.5$], *p* 0.006) and $+5.6\%$ (CI 95% [$-4.7$, 15.9], *p* 0.29) mean knowledge test score, respectively, compared to participants with high VSA (**A**). Participants with low or medium VSA that used an iM2D model had a $-3.7\%$ (CI 95% [$-13.8$, 6.3], *p* 0.47) and $-0.3\%$ (CI 95% [$-10.6$, 10.0], *p* 0.96) mean knowledge test score, respectively, compared to participants with high VSA (**B**).

**Table 2.** Statements related to the visualizations of the anatomical model, grouped by VSA.

| Statement | Mean (±SD) | n | VSA |
|---|---|---|---|
| The 3D model allowed me to better understand spatial * relationships between anatomical structures than 2D views alone. | 4.4 (±0.7) | 22 | high |
| | 4.2 (±0.7) | 22 | medium |
| | 4.3 (±0.9) | 25 | low |
| The 3D model allowed me to better understand the location of organs inside the animal as a whole. | 4.3 (±0.7) | 22 | high |
| | 3.9 (±0.9) | 22 | medium |
| | 4.2 (±0.9) | 24 | low |
| I found it useful to be able to rotate and see the model from different viewpoints. | 4.7 (±0.5) | 22 | high |
| | 4.5 (±0.5) | 22 | medium |
| | 4.5 (±0.7) | 24 | low |

Response options for the above statements were defined on a 5-point Likert scale: 1 = strongly disagree; 2 = disagree; 3 = neither agree nor disagree; 4 = agree; 5 = strongly agree. * spatial relationships were defined as how anatomical structures are located in the body relative to each other; SD = standard deviation; VSA = visual-spatial ability.

## 4. Discussion

In this study, we show that students with low VSA significantly underperform compared to participants with medium and high VSA when using iM3D visualization to study anatomy. When VSA was not taken into consideration, students performed equally well with both the iM2D and iM3D visualizations. Participants with low VSA performed equally as well as those in the medium and high VSA categories when presented with an iM2D model. In addition, participants self-reported the advantages of using iM3D visualizations, regardless of their measured performance.

VSA and the visualization modality both had a modifying effect on the learning outcome, being spatial anatomical knowledge, which means that the VSA and visualization modalities are crucial factors that significantly influence the learning outcomes and cannot be taken out of the equation when describing these relationships. The stratification of learning outcome by the level of VSA based on the MRT score has been used by others to investigate a possible aptitude treatment interaction [29,52,53]. Our findings that iM3D visualization has a significant negative effect on the learning outcome for students with low VSA are supported by others [30,31,54]. It may be explained by the increased cognitive load due to studying multiple viewpoints of the object and constructing a mental three-dimensional image [30,55,56]. Indeed, in our study, the rotation of the anatomical model in the iM3D visualization modality exposed the students to numerous viewpoints from various angles. When these participants were presented with a non-rotatable model (iM2D visualization), they performed equally well as their peers, which could indicate a better learning environment for them with respect to cognitive load. This is supported by the key view paradigm, described by Garg et al., which describes the relationship between VSA and the efficacy of learning by key views or different perspectives [32].

Although students with low VSA did not benefit from the iM3D visualization, they reported a positive experience. They stated that the rotation of the model was useful and helped them to better understand the location of the organs and spatial relations between anatomical structures. This fits with the often misinterpreted paradigm of students as self-educators who would intuitively know what is best for them [57]. A comprehensive meta-analysis performed by Yammine et al. in 2015 described the effectiveness of three-dimensional visualization technologies (3DVT) in the field of anatomy education and further described significantly better learning outcomes in terms of spatial knowledge with interactive 3D compared to non-interactive 2D digital images in a subgroup analysis of 20 studies [37]. The effect of interaction is also underlined in a previously published meta-analysis, in which we describe that only in interventions that involved active user manipulations was a beneficial effect of stereopsis in 3DVT observed [35]. To avoid confounding effects [58], we compared here the visual presentation of the model (iM2D vs. iM3D) within one level of instructional design by keeping other factors, such as the medium, configuration, and instructional method, constant.

In our study, male students had a significant higher mean score on the MRT test compared to female students. Gender differences in the MRT score in favor of men have been consistently reported in the literature [59,60]. Various biological factors [61], socialization factors [62], or task characteristics [63] are assumed to be responsible for this gender difference. Furthermore, research by Hegarty argues that spatial imagery is not the only component that the MRT test measures and suggests that men are more likely to use more efficient analytic strategies to solve the items on the MRT test [64]. The value of the observed gender difference with respect to VSA in our study is therefore difficult to interpret, other than showing that our study population was representative compared to those in other studies.

Given the observed negative effect of low VSA on learning outcomes with the iM3D visualization in our study, it is meaningful to think of methods that aid students with low VSA in their spatial learning. Some research studies suggest that the addition of orientation references could be helpful to rotate three-dimensional virtual objects and to reduce the mental effort it takes to develop a mental representation, especially for students with

lower VSA [65,66]. Furthermore, there is some evidence that multiple orientations of an object offer only minimal advantage over so-called key views or canonical views for spatial learning [30,31]. By providing students with lower VSA with only the critical key views, one could prevent the mobilization of excessive amounts of mental effort, which could lead to cognitive overload. The far end of the spectrum is represented by a full stereoscopic 3D experience, with a model that can be physically or digitally projected in a Head Mounted Display. Most probably, this helps the learners by preventing the necessity to mentally store excessive amounts of spatial data, thereby preventing cognitive overload [29,52].

As VSA is not a static, but rather a dynamic, ability of individuals, it is interesting to consider ways for students to improve their VSA [67]. In this respect, providing adequate training to students with an initially low VSA could help them to reach the level needed to benefit from using iM3D models for learning. A systematic review by Langlois et al. found evidence for the improvement of visual-spatial abilities in anatomy education [68]. Further research into the long-term development of VSA of (medical) students and how anatomy education plays a part in that seems warranted. This is especially because of the tendency to reduce anatomy teaching in medical curricula [4], which could prove harmful for those students who arguably need VSA development the most to perform during their medical skills training [69,70] and to become proficient medical professionals.

Three-dimensional models have the potential to replace the use of laboratory animals in education. Replacing laboratory animals for training researchers in the life science domain and veterinarians is an important topic of discussion in Europe. Recently, the Dutch universities and medical centers published a document aimed at decreasing laboratory animals in higher education using animal-free training methods [14]. It is extremely relevant to innovate education with methods and technologies that are proven to be beneficial for the student, and to inform students about the learning methods that are most beneficial by taking into account their individual needs and abilities. Aside from the laboratory animal-free ethical quest, 3D models could be of great help to avoid the safety and health problems related to cadavers stored in formalin [6].

*Limitations*

Worth to note is that this study had some limitations. First, the most important difference in our study between the iM2D and the iM3D visualizations was the ability to rotate the anatomical model. However, we did not formally measure if and how much the participants used the rotation function while studying with the iM3D visualization, although it can be expected that the participants used this feature. Consequently, we could only draw conclusions in our study about the effect of adding the possibility for students to use multiple viewpoints in the iM3D visualization. Second, the two study sessions and coupled knowledge tests each covered different organ systems (i.e., digestive and urogenital systems). This was carried out to prevent the obtained knowledge in the first session from influencing the learning outcomes in the second session. A difference in previous knowledge about either organ system could have had an influence on the knowledge test score regardless of the visualization used for studying. This issue could have been addressed by offering a pre-test to evaluate participants' baseline knowledge about the subject. Thirdly, because of COVID-19 restrictions, we allowed students to participate remotely from any location and use their own computer hardware. This probably has resulted in a difference in user experience between participants, which could have had an influence on the results, for example, using a trackpad instead of a computer mouse to interact with the anatomical model or differences in monitor sizes. Lastly, we have not investigated if the time limit given for doing the exercise exacerbates the hypothesized results of lower VSA performance in different visual modalities, and we cannot rule out the possibility of lower-MRT participants reaching the same scores using the iM3D visualization when granted more time.

**Author Contributions:** Conceptualization: B.S.v.L., B.P.H. and D.C.F.S.; data curation: B.S.v.L.; funding acquisition: B.P.H., D.C.F.S. and B.S.v.L.; investigation: B.S.v.L. and A.E.D.D.; formal analysis: B.S.v.L. and A.E.D.D.; methodology: B.S.v.L., B.P.H. and J.C.M.V.; software: B.S.v.L.; writing—original draft: B.S.v.L., A.E.D.D. and B.P.H.; writing—review and editing: B.S.v.L., J.C.M.V., B.P.H. and D.C.F.S. All authors have read and agreed to the published version of the manuscript.

**Funding:** This research was funded by Utrecht Stimuleringsfonds Onderwijs 2021 "Avatars of animals and humans: 3D interactive holographic models", The Erasmus+ project 2020-1-NL01-KA226-HE-083056 "Veterinary Online Collection", and the Proefdiervrij Venture Challenge 2021 "Avatar Zoo" (Supported by Proefdiervrij NL, https://proefdiervrij.nl).

**Institutional Review Board Statement:** This study was conducted in accordance with the Declaration of Helsinki and approved by the Institutional Review Board (or Ethics Committee) of the Netherlands Association of Medical Education (NVMO) (dossier number: 2021.6.9).

**Informed Consent Statement:** Informed consent was obtained from all subjects involved in the study.

**Data Availability Statement:** The data presented in this study are available upon request from the corresponding author. The data are not publicly available due to ethical restrictions.

**Acknowledgments:** We would like to thank the following people: Daniel Jansma (LUMC) supplied the script for the functionality and layout of the webviewer. Jozien Imthorn (LUMC) took care of the logistics and planning related to the LAS course. Tineke Coenen-de Roo (LUMC), Marleen Blom (LUMC), Pim Rooymans (UU), Esther Langen (UU) gave permission for performing the experiments during the LAS courses.

**Conflicts of Interest:** The authors declare no conflict of interest. The funders had no role in the design of the study; in the collection, analyses, or interpretation of data; in the writing of the manuscript; or in the decision to publish the results.

## Appendix A

**Table A1.** Learning goals Digestive System rat.

| Learning Goals Digestive System<br>At the End of the Learning Session, Students Should Be Able to: | |
| --- | --- |
| Identify the following structures: | Esophagus<br>Stomach<br>Duodenum<br>Jejunum<br>Ileum<br>Cecum<br>Colon<br>Liver<br>Spleen<br>Pancreas<br>Caudal vena cava<br>Aorta<br>Cranial mesenteric artery |
| Describe the course of the following structures: | Duodenum<br>Cecum<br>Colon<br>Caudal vena cava<br>Aorta<br>Cranial mesenteric artery |
| Identify the organs that are in direct contact with the following structures (disregarding any possible mesenteric membranes): | Duodenum<br>Cecum<br>Colon<br>Pancreas |

**Table A2.** Learning goals urogenital system of the male rat.

| **Learning Goals Urogenital System of the Male Rat**<br>**At the End of the Learning Session, Students Should Be Able to:** | |
| --- | --- |
| Identify the following anatomical structures: | Testis<br>Epididymis<br>Ductus deferens<br>Prostate<br>Vesicular glands<br>Coagulating glands<br>Penis<br>Kidneys<br>Adrenal glands<br>Ureter<br>Bladder<br>Rectum<br>Pelvic bone |
| Determine the position of the following anatomical structures relative to the floor or bottom of the pelvic bone: | Rectum<br>Bladder<br>Penis<br>Kidneys<br>Testis |
| Identify the structures that are in direct contact with the following organs (disregarding any possible mesenteric membranes): | Vesicular gland<br>Bladder |
| Describe the course of the following anatomical structures and how they run relative to each other: | Ductus deferens<br>Ureter |

## Appendix B

*Appendix B.1. Knowledge Test 1*

1. Which part of the digestive tract enters (1) and which part leaves (2) the caecum? Additionally, on which side of the midline of the abdomen is the cecum located in the rat (3)?

(1)_______________________________________________________________
(2)_______________________________________________________________
(3)_______________________________________________________________
Answer: 1. ileum (1/2 pt.) 2. colon (1/2 pt.) 3. left (1 pt.)

2. The colon runs in a ................ (1) direction where it passes the cranial mesenteric artery to the left/right/cranial/caudal (2) side. Then, it runs in a ................ (3) direction where it passes the cranial mesenteric artery to the left/right/cranial/caudal (4) side. Lastly, it runs in a ................ (5) direction where it passes the cranial mesenteric artery to the left/right/cranial/caudal (6) side.

(1)_______________________________________________________________
(2)_______________________________________________________________
(3)_______________________________________________________________
(4)_______________________________________________________________
(5)_______________________________________________________________
(6)_______________________________________________________________
Answer: 1. cranial (1/3 pt.) 2. right (1/3 pt.) 3. left (1/3 pt.) 4. cranial (1/3 pt.) 5. caudal (1/3 pt.) 6. left (1/3 pt.)

3. Which of these anatomical structures are in direct contact to the pancreas in the rat? (You can select multiple answers)

☐ Stomach

☐ Liver

☐ Caecum

☐ Duodenum

☐ Ileum

☐ Colon

☐ Spleen

☐ Right kidney

Answer: stomach (2/3 point), duodenum (2/3 point), spleen (2/3 point) every other box ticked result in $\frac{1}{2}$ point deduction for the total points awarded for this question (with a minimum of 0 points)

4. Where does the caudal vena cava enter the diaphragm? (1) Does the vena cava runs to the left/right/dorsal/ventral side or does it stay at the same course before entering the diaphragm? (2)

(1)_______________________________________________________________

(2)_______________________________________________________________

Answer: 1. right side of diaphragm (1 pt.). 2. runs ventral (to the right) (1 pt.)

*Appendix B.2. Knowledge Test 2*

1. Describe the position of the bulbourethral glands relative to the penis.

_______________________________________________________________

Answer: dorsally (1 pt.) and at the proximal/beginning part (1 pt.) of the penis

2. Describe the position of the testis (1) and the right and left kidneys (2) relative to the floor of the pelvic bone.

_______________________________________________________________

Answer: testis are located caudal and ventral to the floor of the pelvic bone (1 pt.) The kidneys are located cranial and dorsal to the floor of the pelvic bone. (1 pt.)

3. Describe in which direction the ductus deferens runs and how the ductus deferens crosses the ureter along its way to join the urethra.

_______________________________________________________________

Answer: The ductus deferens loops over the ureter (1 pt.) and turns caudally (1 pt.) (between the bladder and a group of accessory genital glands to join the urethra)

4. Which of these anatomical structures are in direct contact to the bladder in the rat? (multiple answers can be selected)

☐ Prostate

☐ Bulbourethral gland

☐ Ureter

☐ Penis

☐ Rectum

☐ Left kidney

☐ Sacrum

☐ Vesicular glands

Answer: prostate (2/3 point), ureter (2/3 point), Vesicular Glands(2/3 point) every other box ticked result in $\frac{1}{2}$ point deduction for the total points awarded for this question (with a minimum of 0 points)

## Appendix C

*Questionnaire*

|  | Strongly Disagree | Disagree | Neither Agree Nor Disagree | Agree | Strongly Agree |
|---|---|---|---|---|---|
| The 3D model allowed me to better understand spatial * relationships between anatomical structures than 2D views alone. | ○ | ○ | ○ | ○ | ○ |
| The 3D model allowed me to better understand the location of organs inside the animal as a whole. | ○ | ○ | ○ | ○ | ○ |
| I found it useful to be able to rotate and see the model from different viewpoints. | ○ | ○ | ○ | ○ | ○ |

* spatial relationships is defined as how anatomical structures are located in the body relative to each other.

Did you experience any problems or difficulties during the use of the web application?

○ Yes, please clarify:

_______________________________________

○ No

Do you have any other feedback on the experiment?

○ Yes, please clarify:

_______________________________________

○ No

## Appendix D

**Table A3.** Baseline demographic and visual-spatial ability characteristics by academic institutes and by total.

| Characteristic | Leiden University Medical Center (n = 49) | Utrecht University (n = 20) | Total (n = 69) | *p*-Value |
|---|---|---|---|---|
| Gender, n (%) |  |  |  |  |
| Female | 36 (73) | 11 (55) | 47 (68.1) |  |
| Male | 12 (24) | 9 (45) | 21 (40.4) | 0.21 |
| Unknown | 1 (2) | 0 (0) | 1 (1.4) |  |
| Age, mean (±SD) | 26.0 (4.2) | 26.4 (5.8) | 26.1 (4.7) | 0.97 |
| Previous Education, n (%) |  |  |  |  |
| ≤BSc | 23 (47) | 7 (35) | 30 (43.5) |  |
| ≥MSc | 26 (53) | 13 (65) | 39 (56.5) | 0.36 |
| Mental Rotation Test, mean (±SD) | 16.2 (5.4) | 15.5 (5.6) | 16.0 (5.4) | 0.59 |

*p*-values for the difference in distribution of Gender, Previous Education, and between Academic Institutes were calculated using Pearson's $X^2$ tests and for Age and MRT using the Wilcoxon rank sum test with continuity correction. BSc = Bachelor of Science degree; iM3D = interactive monoscopic three-dimensional; iM2D = interactive monoscopic two-dimensional; MRT = mental rotation test; MSc = Master of Science degree; SD = standard deviation.

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
