# Peer review of "Rotation of 3D Anatomy Models Is Associated with Underperformance of Students with Low Visual-Spatial Abilities: A Two-Center Randomized Crossover Trial"

_education, doi:10.3390/educsci13100992_

Round 1
Reviewer 1 Report
Dear Authors,
I read you manuscript with interest and expertise in the area of education in anatomical realms as it related to spatial abilities, attention, 3D projection of specimens, and cognitive load reduction therein. In this reviewer's opinion the context of the paper is appropriate, the methods appear to be solid, and the results are are both subtle but definitely in line with current thinking in the literature. The mechanisms of how this occurs in your paradigm are tempting to speculate but you did a good job sticking to what the data tells you. In general the paper reads extremely well and I had very little to frustrate my reading. When I thought a point was missing, it usually was addressed in the next paragraph. This indicates to me a high degree of reflection on your experimental approach and the data it yields.
General Comments:
Generally, it always "hard" to understand how models are portrayed in digital environments and I think yours in no exception. The screen shots of the model are fine but the viewer cannot see the buttons to "envision" how one might use either model. With the 3D model, I think the large arrow implies the rat is rolling on the screen. To me, it seems like the 2D model is oriented vertically while in the 3D model it is situated horizontally across the screen. It is not clear if this was intentional or simply not considered.
Secondly, you had a statistician's mind write or help with the analysis model and I was impressed with the forethought. Interestingly, and probably a good thing, the interactions were not significant or your results would be much less clear.
Finally, if there was room, your study extends well beyond the context audience. Unless you've been instructed elsewhere to keep it brief, I think the results tell an important story of covert cognitive loads disguised at progress. If 3D tools do not help the mass of students, while lowly 2D do the trick for everyone, new tools need to be systematically scrutinized. You could blow that horn a little louder in my reviewer's opinion.
Specific Comments:
ln 183 and referring to figure 1
Was there any practice allowed with the software before the learning exercises were started? A concern sometimes is the usability of the software and the metacognition of the first time users who first must” learn how to learn” in novel environments. It doesn’t appear noted here as controlled but perhaps it may be alluded to in the discussion? Perhaps it is not an issue at all and the students immediately knew how to use the software?
I see later on ln 217 that the authors attempt to control for the metacognitive component.
ln 185
This is not a criticism but the "rate of learning", that is how many items per minute of time with the model, should be taken into account for the two visualization modalities and for the learning objectives. It appears that the authors have paid attention to equivalent “amounts” of learning such that cognitive loads of the learners are not skewed due to different learning task items.
It may be spoken of in the discussion but would the student population used here be familiar with the pace set up by the experiment? Does, or will, the time limit of learning exercises exacerbate hypothesized results of lower VSA performance in different visual modalities?
VSA level based on their MRT scores (MRT≤13 = low, MRT 14-19 = me- 287 dium, MRT 20-24 = high)
ln 290
An interesting and telling finding that is only found when the MRT iis partioned into into low to high VSA. This has significant history in the literature with performance on less complex tasks, cerebral blood flow, gaze behaviour, and MRT score differences between these groups.
ln 306
Excellent finding that is often overlooked as the educational community races towards finding the new best digital tool. It is compelling because your feedback suggests a lack of awareness of low VSA to depressions in their performance in iM3D. This reviewer suggests your findings expose what we call silently dangerous to student learning because individuals generally like the experience but don’t succeed like others with higher VSA. This may be cognitive load related but the low VSA persons don't seem to be aware of it.
Author Response
Dear reviewer, please see the attachment for our response to your comments and suggestions.

Reviewer 2 Report
This study compared the effectiveness of two interactive visualization methods, iM3D and iM2D, for learning spatial anatomy using virtual 3D models as an alternative to cadaveric dissection. Participants with lower Visual Spatial Ability (VSA) performed worse with iM3D, suggesting that the mental 3D image construction in this method could overload students with limited VSA. Further research could focus on adapting visualization techniques to individual students' needs. The text is well-written, materials and methods described in sufficient detail and the results adequately discussed. In my opinion, this study highlights an important aspect that should be considered when designing the most effective learning tools.
I only have a few minor comments that should be addressed:
L31-35: Some studies also mention other disadvantages of harmful animal use in education, such as distraction from relevant scientific concepts by the plight of the animals (Knight 2011) or psychological impact on students (Capaldo 2004). Educational experiences perceived as morally wrong might lead to desensitization (Arluke and Hafferty 1996) and compassion fatigue (Colombo et al. 2016).
L37-39: “…which legally obliges educators to introduce, whenever possible, alternatives to the educational use of animals.” Unfortunately, this requirement is often not complied with. As demonstrated with the constantly high numbers in the EU statistics of animal use for educational purposes and a study by Zemanova et al. (Zemanova et al. 2021).
L50-51: In fact, several reviews have shown that teaching without harmful animal use leads to equivalent or even better learning outcomes (Patronek and Rauch 2007; Knight 2007; Zemanova and Knight 2021).
L91-96: It would be useful to provide more context about the LAS course in terms of what replacement strategies are already being implemented and with how much success.
L391-393: This sentence is not very clear. Could you elaborate on the VSA development impact and provide some references for your statement?
References:
1. Arluke A, Hafferty F (1996) From apprehension to fascination with ''dog lab'': the use of absolutions by medical students. Journal of Contemporary Ethnography 25 (2):201-225. doi:10.1177/089124196025002002
2. Capaldo T (2004) The psychological effects on students of using animals in ways that they see as ethically, morally or religiously wrong. ATLA 32:525-531. doi:10.1177/026119290403201s85
3. Colombo ES, Pelosi A, Prato-Previde E (2016) Empathy towards animals and belief in animal-human-continuity in Italian veterinary students. Animal Welfare 25 (2):275-286. doi:10.7120/09627286.25.2.275
4. Knight A (2007) The effectiveness of humane teaching methods in veterinary education. ALTEX 24 (2):91-109. doi:10.14573/altex.2007.2.91
5. Knight A (2011) The Costs and Benefits of Animal Experiments. Palgrave Macmillan, Basingstoke, UK
6. Patronek GJ, Rauch A (2007) Systematic review of comparative studies examining alternatives to the harmful use of animals in biomedical education. Journal of the American Veterinary Medical Association 230 (1):37-43. doi:10.2460/javma.230.1.37
7. Zemanova MA, Knight A (2021) The educational efficacy of humane teaching methods: a systematic review of the evidence. Animals 11:114. doi:10.3390/ani11010114
8. Zemanova MA, Knight A, Lybæk S (2021) Educational use of animals in Europe indicates reluctance to implement alternatives. ALTEX 38 (3):490-506. doi:10.14573/altex.2011111
Author Response

(The authors gave the same response as above.)
